# Investigating the Effects of Gossypetin on Liver Health in Diet-Induced Pre-Diabetic Male Sprague Dawley Rats

**DOI:** 10.3390/molecules30081834

**Published:** 2025-04-19

**Authors:** Karishma Naidoo, Andile Khathi

**Affiliations:** Department of Human Physiology, School of Laboratory Medicine and Medical Sciences, College of Health Sciences, University of KwaZulu-Natal, Durban 4000, South Africa; khathia@ukzn.ac.za

**Keywords:** gossypetin, NAFLD, liver injury, flavonoids, steatosis, high-caloric diets

## Abstract

The rising prevalence of non-alcoholic fatty liver disease among patients with type 2 diabetes mellitus has emerged as a global health challenge. Gossypetin (GTIN) is a natural flavonoid which has recently demonstrated antihyperglycaemic, antioxidant, and anti-inflammatory effects. Despite these findings, no studies have investigated its effects on liver health in the pre-diabetic state. Hence, this study aimed to investigate the effects of GTIN on liver health in diet-induced pre-diabetic male rats in the presence and absence of dietary intervention and to compare these effects with those of metformin (MET). Following 20 weeks of pre-diabetes induction, the animals were divided into six groups (*n* = 6) as follows: non-pre-diabetic (NPD) control, pre-diabetic (PD) control, and PD groups treated with GTIN (15 mg/kg body weight (bw)) or metformin (500 mg/kg bw) on either a normal diet or a high-fat, high-carbohydrate diet for 12 weeks. The results showed that the PD group had significantly higher liver triglycerides (TAG), liver weights, sterol regulatory binding element regulatory protein-1c (SREBP-1c), malondialdehyde (MDA) levels, and liver injury enzyme levels, along with decreased liver superoxide dismutase (SOD) activity, glutathione peroxidase (GPx) activity, and plasma bilirubin levels in comparison to NPD. Histologically, there was an increased lipid droplet accumulation and structural disarray in the PD group. GTIN treatment significantly reduced liver TAGs, liver weights, and plasma SREBP-1c levels, as well as improved liver SOD and GPx activity while decreasing liver MDA levels and liver injury enzymes in comparison to the PD control. Notably, GTIN treatment increased plasma bilirubin levels. Liver histology in the GTIN-treated groups revealed decreased lipid droplet accumulation and improved tissue integrity. Similar results were observed for the liver parameters in the MET-treated groups. The findings of this study may suggest that GTIN and MET exhibit therapeutic effects on liver health in diet-induced pre-diabetes in both the presence and absence of diet intervention. Dietary intervention may confer beneficial effects on liver health, with the most favorable therapeutic outcomes observed through a combination of treatment with dietary intervention. Additionally, GTIN may exhibit greater hepatoprotective effects than MET in rats without dietary intervention.

## 1. Introduction

The chronic intake of unhealthy diets coupled with sedentary lifestyles contribute to the development of diabetes and associated liver complications such as non-alcoholic fatty liver disease (NAFLD) [1]. Pre-diabetes precedes the onset of overt type 2 diabetes mellitus (T2DM), with previous studies showing that around 70% of untreated pre-diabetic individuals progress to T2DM [2,3]. Insulin resistance and hyperinsulinemia associated with pre-diabetes contribute to the development of liver complications [4]. Insulin resistance promotes increased liver triglyceride accumulation by disrupting the ability of insulin to suppress adipose tissue lipolysis [5]. Hyperinsulinemia, elevated glucose, and free fatty acid flux into the liver upregulates hepatic de novo lipogenesis through steroid regulatory elementary binding protein-1c [6]. The accumulation of lipids in the liver contributes to hepatic insulin resistance and the development of NAFLD [7]. This triggers a cascade of oxidative and inflammatory events which leads to liver damage [7]. Moreover, it has been shown that liver injury marker levels which include alanine transaminase and aspartate transaminase were significantly higher in the pre-diabetic state [8]. Histopathological changes observed in NAFLD include lipid droplet accumulation, inflammation, and lobular disarray [9]. This condition may not only disrupt the normal functioning of the liver but can also create a feedback loop in which liver impairment further exacerbates the pre-diabetic state [10].

The main approaches to the management of diabetes-associated liver dysfunction include a combination of dietary and pharmacological intervention [11,12]. There has been low patient compliance with lifestyle modifications due to the challenges of maintaining long-term dietary change and exercise [13,14]. Metformin is primarily used as a first-line treatment for T2DM and has been shown to improve liver health in the diabetic state [15]. However, there have been some challenges with the current drugs under investigation, which include inconsistent findings and a lack of evidence supporting their effectiveness in humans [16]. Thus, there is a need to identify compounds which are effective in treating diabetic liver complications in the presence and absence of dietary modifications. Flavonoids found in plant sources have been shown to improve T2DM and associated liver complications through their multifactorial effects, low toxicity, and vast availability [17,18]. Gossypetin (GTIN) is a naturally occurring flavonoid that serves as a structural analog of quercetin with an additional hydroxyl group [19]. This compound has been isolated from the calyx and flowers of *Hibiscus sabdariffa* [20]. Previous studies have highlighted GTIN for its potent antioxidant and anti-inflammatory properties [21,22]. Recent studies have demonstrated that GTIN improves glucose homeostasis and cardiovascular function in the pre-diabetic state [23,24]. Despite these current findings, no studies have explored the effects of GTIN on liver health in the pre-diabetic state. Hence, this study aimed to investigate the effects of GTIN on liver health in both the presence and absence of dietary intervention in a diet-induced pre-diabetic male rat model and to compare these effects with those of metformin. The oral dose of 15 mg/kg GTIN was selected to be used in our study based on previous studies, demonstrating its efficacy and safety within the 10–20 mg/kg range [25,26]. The 500 mg/kg dose of MET was selected due to its established therapeutic benefits and minimal adverse effects [4,27].

## 2. Results

### 2.1. Liver Triglycerides and Liver Weights

Liver triglyceride levels and liver weights were measured to assess the extent of lipid accumulation and liver enlargement. A one-way ANOVA showed significant differences in liver TAG levels (F (5, 30) = 68.11, *p* < 0.0001) and liver weights (F (5, 30) = 78.87, *p* < 0.0001) between the groups. Post hoc comparisons showed (Figure 1) that the pre-diabetic (PD) control group exhibited significantly (*p* < 0.05) higher liver TAG levels and liver weights in comparison to the non-pre-diabetic (NPD) group. However, the PD animals receiving gossypetin with a normal diet (GTIN+ND) and a high-fat, high-carbohydrate diet (GTIN+HFHC) exhibited significantly (*p* < 0.05) lower liver TAG levels and liver weights in comparison to the PD control. Similar results (*p* < 0.05) were observed in the PD animals receiving metformin with a normal diet (MET+ND) and high-fat, high-carbohydrate diet (MET+HFHC) for liver TAG levels and liver weights. A two-way ANOVA revealed that both treatment (F (1, 20) = 45.16, *p* < 0.0001) and diet (F (1, 20) = 67.03, *p* < 0.0001) exhibited significant main effects on liver TAGs. Similarly, significant main effects of treatment (F (1, 20) = 18.08, *p* < 0.0004) and diet (F (1, 20) = 79.37, *p* < 0.0001) were observed for liver weight. There was a significant interaction between treatment and diet on liver TAG (F (1, 20) = 10.73, *p* = 0.0038). However, there was no significant interaction between treatment and diet on liver weight (F (1, 20) = 3.76, *p* = 0.0667). Post hoc comparisons showed that GTIN treatment (GTIN+HFHC) resulted in significantly (*p* < 0.05) lower liver TAG levels and liver weights in comparison to MET treatment (MET+HFHC). Animals on an ND (GTIN+ND and MET+ND) exhibited significantly (*p* < 0.05) lower levels of both parameters in comparison to those on a HFHC diet (GTIN+HFHC and MET+HFHC).

### 2.2. Plasma SREBP-1c Levels

Plasma SREBP-1c levels were measured to assess the regulation of lipogenesis in the liver. A one-way ANOVA showed significant differences (F (5, 30) = 74.47, *p* < 0.0001) in plasma SREBP-1c levels between the groups. Post hoc comparisons showed (Figure 2) that the PD control group exhibited significantly (*p* < 0.05) higher plasma SREBP-1c levels in comparison to the NPD group. However, the GTIN+ND and GTIN+HFHC groups exhibited significantly (*p* < 0.05) lower plasma SREBP-1c levels in comparison to the PD control. Similar results (*p* < 0.05) were observed for plasma SREBP-1c levels in the MET+ND and MET+HFHC groups. A two-way ANOVA revealed that both treatment (F (1, 20) = 24.86, *p* < 0.0001) and diet (F (1, 20) = 223.30, *p* < 0.0001) exhibited significant main effects on plasma SREBP-1c levels. There was a significant (F (1, 20) = 10.74, *p* = 0.0038) interaction between treatment and diet on plasma SREBP-1c levels. Post hoc comparisons showed that GTIN treatment (GTIN+ND) resulted in significantly (*p* < 0.05) lower plasma SREBP-1c levels in comparison to MET treatment (MET+ND). Animals on an ND (GTIN+ND and MET+ND) exhibited significantly (*p* < 0.05) lower levels of this parameter in comparison to those on a HFHC diet (GTIN+HFHC and MET+HFHC).

### 2.3. Liver Oxidative Status

Liver oxidative status was assessed by measuring MDA levels for lipid peroxidation and SOD and GPx activities for enzymatic antioxidant defense. A one-way ANOVA showed significant differences in liver MDA (F (5, 30) = 14.44, *p* < 0.0001), SOD activity (F (5, 30) = 211.50, *p* < 0.0001), and Gpx activity (F (5, 30) = 29.17, *p* < 0.0001) between the groups. Post hoc comparisons showed (Figure 3) that the PD control group exhibited significantly (*p* < 0.05) higher liver MDA levels with reduced SOD and GPx activity in comparison to the NPD group. However, the GTIN+ND and GTIN+HFHC groups exhibited significantly (*p* < 0.05) lower liver MDA levels with improved SOD and GPx activity in comparison to the PD control. Similar results (*p* < 0.05) were observed for liver MDA levels and Gpx activity in the MET+ND and MET+HFHC groups. The MET+ND group showed significantly (*p* < 0.05) higher liver Gpx activity in comparison to the PD control.

A two-way ANOVA revealed that both treatment (F (1, 20) = 10.51, *p* < 0.0041) and diet (F (1, 20) = 34.04, *p* < 0.0001) exhibited significant main effects on liver MDA levels. Similarly, both treatment (F (1, 20) = 252.90, *p* < 0.0001) and diet (F (1, 20) = 138.50, *p* < 0.0001) exhibited significant main effects on liver SOD activity. Furthermore, both treatment (F (1, 20) = 10.59, *p* = 0.0040) and diet (F (1, 20) = 21.54, *p* = 0.0002) exhibited significant main effects on liver Gpx activity. There was no significant (*F (1, 20)* = 0.42, *p* = 0.5242) interaction between treatment and diet on liver MDA levels. However, there was a significant interaction between treatment and diet on liver SOD activity (F (1, 20) = 1.05, *p* = 0.3174) and Gpx activity (F (1, 20) = 4.54, *p* = 0.0457). Post hoc comparisons showed that GTIN treatment (GTIN+ND and GTIN+HFHC) resulted in significantly (*p* < 0.05) higher liver SOD activity in comparison to MET treatment (MET+ND and MET+HFHC). Additionally, GTIN treatment (GTIN+HFHC) resulted in significantly (*p* < 0.05) higher liver Gpx activity in comparison to MET treatment (MET+HFHC). Animals on an ND (GTIN+ND and MET+ND) exhibited significantly (*p* < 0.05) lower liver MDA levels and higher SOD activity in comparison to those on a HFHC diet (GTIN+HFHC and MET+HFHC).

### 2.4. Plasma Liver Injury Enzymes

Plasma aspartate transaminase (AST) and alanine transaminase (ALT) were measured to assess liver function and injury. A one-way ANOVA showed significant differences in plasma AST (F (5, 30) = 59.97, *p* < 0.0001) and ALT (F (5, 30) = 122.50, *p* < 0.0001) levels between the groups. Post hoc comparisons showed (Figure 4) that the PD control group exhibited significantly (*p* < 0.05) higher plasma AST and ALT levels in comparison to the NPD group. However, the GTIN+ND and GTIN+HFHC groups exhibited significantly (*p* < 0.05) lower plasma AST and ALT levels in comparison to the PD control. Similar results (*p* < 0.05) were observed for plasma AST and ALT levels in the MET+ND and MET+HFHC groups. A two-way ANOVA revealed that both treatment (F (1, 20) = 33.80, *p* < 0.0001) and diet (F (1, 20) = 105.60, *p* < 0.0001) exhibited significant main effects on plasma AST levels. Similarly, both treatment (F (1, 20) = 27.82, *p* < 0.0001) and diet (F (1, 20) = 79.67, *p* < 0.0001) exhibited significant main effects on plasma ALT levels. There was a significant (F (1, 20) = 14.33, *p* = 0.0012) interaction between treatment and diet on plasma AST levels. However, there was no significant (F (1, 20) = 3.33, *p* = 0.0829) interaction between treatment and diet on plasma ALT levels. Post hoc comparisons showed that GTIN treatment (GTIN+HFHC) resulted in significantly (*p* < 0.05) lower plasma AST and ALT levels in comparison to MET treatment (MET+HFHC). Animals on an ND (GTIN+ND and MET+ND) exhibited significantly (*p* < 0.05) lower levels of both these parameters in comparison to those on a HFHC diet (GTIN+HFHC and MET+HFHC).

### 2.5. Plasma Bilirubin

Plasma bilirubin levels were measured to assess liver function and antioxidant status in the pre-diabetic state. A one-way ANOVA showed significant (F (5, 30) = 62.93, *p* < 0.0001) differences in plasma bilirubin levels between the groups. Post hoc comparisons showed (Figure 5) that the PD control group exhibited significantly (*p* < 0.05) lower plasma bilirubin levels in comparison to the NPD group. However, the GTIN+ND and GTIN+HFHC groups exhibited significantly (*p* < 0.05) higher plasma bilirubin in comparison to the PD control. Similar results (*p* < 0.05) were observed for plasma bilirubin levels in the MET+ND and MET+HFHC groups. A two-way ANOVA revealed that both treatment (F (1, 20) = 14.85, *p* < 0.0010) and diet (F (1, 20) = 51.21, *p* < 0.0001) exhibited significant main effects on plasma bilirubin levels. There was a significant (F (1, 20) = 4.85, *p* = 0.0396) interaction between treatment and diet on plasma bilirubin levels. Post hoc comparisons showed that GTIN treatment (GTIN+HFHC) resulted in significantly (*p* < 0.05) higher plasma bilirubin levels in comparison to MET treatment (MET+HFHC). Animals on an ND (GTIN+ND and MET+ND) exhibited significantly (*p* < 0.05) higher levels of this parameter in comparison to those on a HFHC diet (GTIN+HFHC and MET+HFHC).

### 2.6. Liver Histology

Liver histology was measured to assess liver architecture and cellular damage. The results (Figure 6B) showed increased lipid droplet accumulation, hepatocyte nucleus displacement, disturbed sinusoidal space, and hepatocyte plate arrangement in the PD group in comparison to the NPD group. There was no remarkable difference in the number of binucleated hepatocytes in the PD group in comparison to the NPD group. The GTIN+ND (Figure 6C) and GTIN+HFHC (Figure 6D) groups showed a decrease in lipid droplet accumulation and hepatocyte nucleus displacement, as well as improved sinusoidal space and hepatocyte arrangement in comparison to the PD group. Similar results were observed in the MET+ND (Figure 6E) and MET+HFHC (Figure 6F). There was no remarkable difference in the number of binucleated hepatocytes in the treatment groups (GTIN+ND, GTIN+HFHC, MET+ND, MET+HFHC) in comparison to the PD group. The histopathological scores for each parameter are shown in Table 1. For each histological parameter, the scores were the same for all animals within each group.

## 3. Discussion

Excess glucose influx into the liver triggers the activation of lipogenic transcription factor SREBP-1c, which drives de novo lipogenesis and leads to increased TAG accumulation [28]. In this study, the liver TAG levels, liver weights, and plasma SREBP-1c levels were significantly higher in the PD group in comparison to the NPD group. The development of NAFLD has been associated with unhealthy dietary patterns characterized by high saturated fat and carbohydrate content along with a low intake of fruits and vegetables [29]. Additionally, fructose has been identified as a key factor in the development of NAFLD in humans and rodents [30]. The high-fat, high-carbohydrate (HFHC) diet and fructose-supplemented water used to induce pre-diabetes in our study may have promoted insulin resistance contributing to hyperinsulinemia and elevated glucose levels, as previously reported [31]. This may have led to increased liver TAG accumulation through increased adipose tissue lipolysis and hepatic lipogenesis via SREBP-1c in the PD group. The increased liver weights may have been attributed to the accumulation of TAGs in the liver, which is a key feature of NAFLD.

However, both GTIN-treated groups exhibited significantly lower liver TAG levels, liver weights, and plasma SREBP-1c levels in comparison to the PD control. A recent study showed that the administration of 15 mg/kg body weight (bw) of GTIN promoted weight loss and improved insulin sensitivity by reducing ghrelin levels and caloric intake [23]. Additionally, another study found that the administration of 20 mg/kg bw of GTIN to mice on a methionine–choline-deficient diet for 4 weeks reduced hepatic lipid accumulation by inhibiting lipogenic pathways and promoting β-oxidation [25]. They also found that GTIN attenuates hepatic lipid accumulation by promoting AMP-activated protein kinase (AMPK) activation [25]. Therefore, in our study, we speculate that GTIN may have reduced liver TAG accumulation by improving insulin sensitivity, promoting β-oxidation, and inhibiting lipogenic pathways, as evidenced by the low plasma SREBP1c levels in the GTIN-treated groups. The decreased liver TAG levels may have led to a decrease in liver weight. Similar results were observed for liver TAG levels, liver weights, and plasma SREBP-1c levels in MET-treated groups. Studies have shown that MET was effective in reducing intrahepatic lipid accumulation [32,33]. In skeletal muscle, MET has been reported to activate AMPK, which increases glucose transporter-4 translocation and improves glucose uptake [34,35]. This has shown to reduce insulin resistance, ultimately leading to a reduction in SREBP-1c activation, hepatic steatosis, and liver weight [36].

Animals maintained on a normal diet exhibited significantly lower levels of liver TAGs, liver weights, and plasma SREBP-1c levels in comparison to those on the HFHC diet. Our findings corroborated a previous study which showed that the consumption of a HFHC diet consisting of 30% fat and 55% carbohydrate for 10 weeks promoted increased de novo lipogenesis [37]. Another study showed that mice fed a high-fat diet (HFD) composed of 42% fat and 42.7% carbohydrates for 3 months significantly increased liver weight [38]. Additionally, a study showed that the consumption of a HFD composed of 60% fats for 2 weeks significantly increased liver TAG levels [39]. A significant interaction between treatment and diet was observed for liver TAG levels and plasma SREBP-1c concentrations, whereas no significant interaction was observed for liver weight. These findings may suggest that the therapeutic effects of treatment on liver TAG and SREBP-1c levels may be enhanced when combined with dietary intervention. Furthermore, GTIN treatment exhibited significantly lower liver TAG levels and liver weights in comparison to MET in HFHC-fed animals. GTIN treatment also exhibited significantly lower plasma SREBP-1c levels than MET in animals on a normal diet. These findings may suggest that GTIN exerts greater therapeutic effects on liver TAGs, liver weights, and SREBP-1c levels than MET treatment, particularly in animals without dietary intervention.

Reactive oxygen species (ROS) are generated from various endogenous and exogenous sources, including mitochondrial respiration, enzymatic reactions, and environmental factors [40]. The body utilizes enzymatic and non-enzymatic antioxidant systems to counteract the harmful effects of ROS production [40]. Enzymatic antioxidants include SOD, GPx, and catalase, while non-enzymatic antioxidants include glutathione, vitamin C, vitamin E, and bilirubin, among others [40]. In the pre-diabetic state, liver MDA levels, which are a by-product of lipid peroxidation, were significantly higher in comparison to the control group [41,42]. Insulin resistance associated with NAFLD increases the flow of free fatty acids (FFAs) into the liver which triggers β-oxidation to manage the excess lipids [43]. Mitochondria adapt to lipid accumulation in hepatocytes by increasing β-oxidation [40]. However, increased substrate delivery to the electron transport chain (ETC) leads to increased ROS production [40]. In NAFLD, elevated ROS levels not only promote lipid peroxidation but also deplete antioxidant molecules and impair antioxidant enzyme activity [44]. Patients with NAFLD exhibit a diminished antioxidant capacity in hepatocytes, as evidenced by a significant decrease in enzymatic antioxidants such as SOD, GPx, and catalase [44]. Moreover, oxidative stress plays a pivotal role in the progression from simple steatosis to NASH [45]. In this study, the liver MDA levels were significantly higher in the PD group in comparison to the NPD group. This was further accompanied by significantly lower liver SOD and Gpx activity. This may suggest increased hepatic lipid peroxidation and decreased enzymatic antioxidant activity in the liver during the pre-diabetic state.

However, both GTIN-treated groups showed significantly lower liver MDA levels in comparison to the PD control. This was further accompanied by significantly increased liver SOD and Gpx activity. Flavonoids are natural antioxidants present in the human diet and have gained attention as promising treatment options for NAFLD due to their potent antioxidant effects and safety profile [46,47]. Their therapeutic effects on oxidative stress have been attributed to the activation of AMPK signaling and the nuclear factor erythroid 2-related factor 2 (Nrf2) pathways [48]. AMPK signaling reduces ROS production, while Nrf2 activation promotes the upregulation of GPx and catalase [49]. A recent study showed that the administration of 20 mg/kg bw of GTIN for 4 weeks directly reduces oxidative stress in the liver rather than upregulating Nrf2-dependent antioxidant responses [25]. However, another study showed that the oral administration of 30 mg/kg bw of GTIN for 3 days upregulated SOD expression by increasing the nuclear translocation of Nrf2 [50]. The observed improvement on liver SOD and GPx activity in the GTIN-treated groups of our study may have been due to the longer treatment duration of 12 weeks as compared to the 4-week treatment period reported in the previous study [25]. Similar results were observed for liver MDA levels and SOD activity in the MET-treated groups, while the MET+ND showed comparable Gpx activity. Our study’s results corroborated previous studies showing that MET alleviates oxidative stress in the liver by enhancing the activity of SOD and GPx in a high-fat diet-fed rat model [51,52].

Animals maintained on a normal diet exhibited significantly lower levels of liver MDA as well as higher activities of SOD and GPx in comparison to those on the HFHC diet. The results of this study corroborated a previous study which showed that consumption of a diet containing 44.3% fat and 20% fructose for 4 weeks resulted in increased liver MDA levels and decreased SOD and Gpx activity [53]. Interestingly, a significant interaction between diet and treatment was observed for liver Gpx activities despite small differences in mean values. There was no significant interaction detected for liver MDA and SOD activity. These findings suggest that the most pronounced reductions in liver Gpx activities may be achieved through a combination of pharmacological and dietary intervention. The lack of interaction between treatment and diet on liver MDA levels and SOD activity may suggest that the effects of treatment are not influenced by diet. Furthermore, GTIN treatment demonstrated significantly higher SOD activity in comparison to MET in rats fed a normal diet and HFHC diet. Additionally, GTIN exhibited significantly higher GPx activity in comparison to MET in HFHC-fed animals. This may suggest that GTIN treatment exerts greater beneficial effects than MET on liver SOD and GPx activity in rats without dietary intervention.

Under normal physiological conditions, ALT and AST are enzymes found in hepatocytes [54]. The increased levels of AST and ALT in circulation serve as a marker for liver injury, since these enzymes are released when hepatocytes are damaged [54]. Studies have shown increased plasma ALT and AST levels in the progression of NAFLD [55,56]. Liver injury is driven by insulin resistance, oxidative stress, and inflammation, all of which have been associated with high-caloric diets [57]. Hepatic fatty acid accumulation during insulin resistance enhances β-oxidation, which increases ROS production and promotes oxidative stress [58]. This damages cell membranes of hepatocytes, which leads to liver tissue injury [58]. In this study, the plasma AST and ALT levels were significantly higher in the PD control group in comparison to the NPD group. This may suggest that diet-induced pre-diabetes damages hepatocytes, and this may have been due to hepatic FFA accumulation and lipid peroxidation, as seen in the PD group. However, both GTIN-treated groups showed significantly lower plasma AST and ALT levels in comparison to the PD control. Our results corroborated two recent studies which showed that GTIN administration reduced liver injury [25,50]. In our study, GTIN may have reduced liver damage due to its ability to regulate lipid metabolism and mitigate oxidative stress [25]. Similar results were observed for plasma AST and ALT levels in the MET-treated groups. Previous studies have shown that MET reduces liver injury by enhancing insulin sensitivity and decreasing hepatic lipid accumulation through activation of the AMPK pathway [15,59].

Animals maintained on a normal diet exhibited significantly lower levels of plasma AST and ALT levels in comparison to those on the HFHC diet. Our findings corroborated a previous study which has shown that consumption of a HFD composed of 49% fat for 8 weeks promoted significantly elevated plasma AST and ALT levels [60]. Another study showed that the consumption of a high-carbohydrate diet composed of 80% carbohydrates also promoted an increase in plasma AST and ALT levels [61]. A significant interaction between diet and treatment was observed for plasma AST levels, whereas no significant interaction was detected for plasma ALT levels. These findings may suggest that dietary intervention enhances the beneficial effects of treatment on plasma AST levels. The lack of interaction between treatment and diet on plasma ALT may suggest that the effects of treatment are not influenced by diet. Furthermore, GTIN treatment exhibited significantly lower plasma AST and ALT levels in comparison to MET in HFHC-fed animals. This may suggest that GTIN treatment exerts greater beneficial effects than MET on plasma AST and ALT levels in rats without dietary intervention.

Bilirubin is formed in the reticuloendothelial cells of the liver, spleen, and bone marrow through the breakdown of heme [62]. Studies have shown that bilirubin exerts potent antioxidant, anti-inflammatory, and cytoprotective effects in T2DM, hepatic steatosis, and cardiovascular diseases [63,64]. In this study, the plasma bilirubin levels were significantly lower in the PD control in comparison to the NPD group. This study’s results corroborated previous studies which have shown similar results in the pre-diabetic state [4,65]. Plasma bilirubin levels have been shown to be inversely associated with T2DM and NAFLD [66,67]. Bilirubin has been explored for its therapeutic effects in managing NAFLD in T2DM due to a lack of approved pharmacological treatments [68]. Bilirubin has been shown to enhance insulin sensitivity in diet-induced obese mice by suppressing endoplasmic reticulum stress and chronic inflammation in adipose tissue and the liver [69]. Furthermore, a previous study showed that bilirubin was more effective in protecting against lipid peroxidation than glutathione [70]. Notably, plasma bilirubin contributes to the total antioxidant capacity of the blood [71]. In this study, both GTIN-treated groups exhibited significantly higher plasma bilirubin levels in comparison to the PD control. The increased plasma bilirubin may have contributed to the reduction in hepatic oxidative stress. These findings suggest that GTIN may contribute to improved liver health by enhancing antioxidant plasma bilirubin levels. Similar results were observed for plasma bilirubin levels in the MET-treated groups. This study’s results corroborated a previous study showing elevated plasma bilirubin levels after MET administration [65]. Notably, animals maintained on a normal diet exhibited significantly higher plasma bilirubin levels in comparison to those on the HFHC diet. Our findings corroborated a previous study which has shown that plasma bilirubin decreased after consumption of a HFD [72]. A significant interaction between diet and treatment was observed for plasma bilirubin levels. This may suggest that the effects of pharmacological treatment on plasma bilirubin levels may be enhanced when combined with dietary intervention. Furthermore, GTIN treatment exhibited significantly higher plasma bilirubin levels in comparison to MET in HFHC-fed animals. This may suggest that GTIN treatment exerts greater beneficial effects than MET on plasma bilirubin in rats without dietary intervention.

Under normal physiological conditions, the liver is composed of hexagonal lobules with hepatocytes arranged in radial plates around the central vein [73]. Sinusoids are located between hepatocytes and facilitate the exchange of nutrients and waste [73]. Macrovesicular steatosis is a key histological feature of NAFLD and has been observed in the pre-diabetic state [65,74]. The accumulation of fat in hepatocytes displaces the nucleus to the periphery and lipid-laden hepatocytes reduce sinusoidal space in NAFLD [75]. High-caloric diets and insulin resistance promotes increased hepatic FFA influx, which contributes to hepatic steatosis [43]. In this study, the PD group exhibited notable features such as increased lipid droplet accumulation, the displacement of hepatocyte nuclei, disturbed sinusoidal space, and hepatocyte plate arrangement in comparison to the NPD group. The increased intrahepatic TAG accumulation seen in the PD group may have contributed to the macrovesicular steatosis present histologically. Furthermore, the hepatocyte plate arrangement may have been disrupted due to fat deposition, leading to disarray within the liver lobules. However, both GTIN-treated groups showed reductions in lipid droplet accumulation, displaced hepatocyte nuclei, and improved sinusoidal space and hepatocyte arrangement in comparison to the PD group. These improvements may have been attributed to the antilipidaemic, antioxidant, and anti-inflammatory properties of GTIN [21,22]. Additionally, the MET+ND and MET+HFHC groups showed similar histological improvements as the GTIN-treated groups. However, the MET+HFHC group showed increased lipid droplet accumulation, reduced sinusoidal space, and moderate disruption to hepatocyte plate arrangement in comparison to the MET+ND group. The HFHC diet may have limited the ability of MET to effectively reduce hepatic lipid accumulation [5]. There was no remarkable difference in the number of binucleated hepatocytes among all six groups, which may suggest that hepatocyte regeneration may be occurring at similar rates.

This study highlights the potential of GTIN as an effective compound for ameliorating liver complications associated with the pre-diabetic state. The results demonstrated that GTIN administration in the presence and absence of dietary intervention significantly improved liver TAG levels, liver weights, plasma SREBP-1c, and bilirubin levels in comparison to the PD group. Additionally, GTIN administration reduced liver MDA levels and increased liver SOD and Gpx activity. It also decreased liver injury, demonstrated by reduced plasma AST and ALT levels. Moreover, GTIN improved liver histology by reducing steatosis and restoring liver architecture in comparison to the PD group. The effects of GTIN on liver parameters were comparable to MET treatment. Moreover, dietary intervention may positively influence liver health in pre-diabetes. However, the combination of pharmacological and dietary intervention may exhibit greater therapeutic outcomes on liver health than when used independently. Notably, GTIN may exert greater protective effects than MET in rats without dietary intervention.

## 4. Materials and Methods

### 4.1. Chemicals and Drugs

The chemicals employed were of analytical grade and sourced from standard commercial suppliers (Merck chemicals (PTY) Ltd., Wadeville, Gauteng, South Africa). Gossypetin was purchased from INDOFINE Chemical Company (Hillsborough, NJ, USA). Metformin was purchased from Sigma-Aldrich (St. Louis, MI, USA).

### 4.2. Animals and Housing

The study involved 36 male Sprague Dawley rats (150–180 g), which were bred and kept in the Biomedical Research Unit (BRU) at the University of KwaZulu-Natal (UKZN), Westville campus. Under standard laboratory conditions, the animals were housed at a constant temperature of 22 ± 2 °C, a carbon dioxide (CO_2_) content of <5000 p.m, relative humidity of 55 ± 5%, and a 12 h light/dark cycle (lights on at 07:00). Noise levels were kept under 65 decibels. Food and water were provided ad libitum to all animals. All animal experiments and handling were approved by the Animal Research Ethics Committee of the University of KwaZulu-Natal (ethics number: AREC/0000495/2022). Pain and discomfort were closely monitored in accordance with the humane endpoint criteria outlined by the UKZN Institutional Animal Ethics Committee. The rats were housed in well-ventilated polycarbonate plastic cages (Techniplast, Labotec, Midrand, South Africa). Before the experimental diet (high fat high carbohydrate) was introduced, the rats were allowed to acclimatize to their new environment while consuming a standard rat diet (Meadow Feeds, Lanseria, South Africa) and tap water for 1 week [31].

### 4.3. Induction of Pre-Diabetes

The animals were randomly distributed into two groups based on diet (Figure 7) as follows: group 1 (*n* = 6) and group 2 (*n* = 30). The induction of experimental pre-diabetes in the animals was carried out using the protocol outlined by Luvuno et al. [31]. Group 1 animals were given a standard rat diet and tap water, while group 2 animals received a high-fat, high-carbohydrate (HFHC) diet (AVI Products (Pty) Ltd., Waterfall, Johannesburg, South Africa) supplemented with 15% fructose-enriched water to induce pre-diabetes for 20 weeks. The HFHC diet was composed of 55% carbohydrates, 30% fats, and 15% proteins of the total energy content [31]. Following 20 weeks, the animals were assessed for pre-diabetes using the American Diabetes Association criteria [76]. Animals with a fasting blood glucose concentration of 5.6 to 6.9 mmol/L and a 2 h oral glucose tolerance test glucose concentration of 7.8 to 11.0 mmol/L were classified as the pre-diabetic group. The animals on the standard diet were also assessed at week 20 to confirm normoglycemia and were categorized as the non-pre-diabetic group. To measure glucose concentration, blood was collected using the tail-prick method [77]. Glucose levels were assessed using a OneTouch Select glucometer (LifeScan, Mosta, Malta). Blood samples were taken at week 20 and every 4 weeks during the treatment period to monitor changes in glucose levels. The sample size was selected based on similar studies and power analysis to ensure adequate statistical power while ensuring animal welfare [65,78].

### 4.4. Experimental Design and Treatment

Following the induction of pre-diabetes, the pre-diabetic group (*n* = 30) was randomly subdivided into five groups (Groups 2 to 6), each group comprising six animals each. The pre-diabetic animals either continued the HFHC diet or changed to a normal standard diet (ND) while receiving either 15 mg/kg body weight (b.w) gossypetin (GTIN) or 500 mg/kg bw metformin (MET) orally once every third day at 9:00 a.m. for 12 weeks. The doses of 15 mg/kg GTIN and 500 mg/kg MET were selected based on previous studies, demonstrating their effectiveness in similar pre-diabetic rat models [4,23]. The non-pre-diabetic control (NPD, Group 1) animals were continuously fed on a normal diet (ND). The pre-diabetic control (PD, Group 2) group consisted of untreated pre-diabetic rats that were continuously fed a HFHC diet. Group 3 (GTIN+ND) animals were changed to the ND and received GTIN, whereas Group 4 (GTIN+HFHC) animals remained on the HFHC diet and received GTIN. Group 5 (MET+ND) animals were changed to the ND and were given MET, while Group 6 (MET+HFHC) animals continued on the HFHC diet and received MET. In all treatment groups, the drugs were dissolved in 3 mL/kg of diluted dimethyl sulfoxide (DMSO) (prepared as 2 mL DMSO in 19 mL normal saline, p.o). To ensure consistency and control for any potential effects of the vehicle, the control groups were also administered the same dose of DMSO without any dissolved drug. We have used 10% DMSO as a vehicle, as it has proven to dissolve organic compounds based on previous studies in our laboratory [79,80]. The amount of DMSO administered was 3 mL/kg every third day. We have selected this volume based on the recommended safe volume to be administered to rodents [81].

### 4.5. Blood Collection and Tissue Harvesting

At the end of week 32 of the experimental period, all animals were anesthetized for 3 min with Isofor (100 mg/kg bw) (Safeline Pharmaceuticals (Pty) Ltd., Roodeport, South Africa) using a gas anesthetic chamber (BRU, UKZN, Durban, South Africa). After the animals were anesthetized, blood was obtained via cardiac puncture and placed into pre-cooled heparinized containers. The blood samples were centrifuged (Eppendorf centrifuge 5403, Germany) at 4 °C, at 503× *g* for 15 min to obtain plasma. Each plasma sample was transferred into Eppendorf tubes and stored in a Bio Ultra freezer (Snijers Scientific, Tilburg, The Netherlands) until ready for biochemical analysis. The liver was removed, weighed, and rinsed with a cold normal saline solution. The left lobe was immediately snap-frozen in liquid nitrogen and stored at −80 °C in a BioUltra freezer (Snijers Scientific, Tilburg, The Netherlands) for biochemical analysis, while the right lobe was preserved in 10% formalin buffer for histopathological examination.

### 4.6. Biochemical Analysis

At the end of week 32 of the experimental period, various biochemical parameters were assessed.

#### 4.6.1. Plasma Biochemical Analysis

Plasma aspartate transaminase (AST), alanine transaminase (ALT), and bilirubin levels were analyzed with IDEXX Catalyst One Chemistry Analyzer (IDEXX Laboratories Inc. Westbrook, ME, USA). Plasma sterol regulatory element binding protein 1c (SREBP-1c) levels were measured using the relevant ELISA kit (catalog no.: EKC39725) according to the manufacturer’s instructions (Biomatik USA, LLC, Wilmington, DE, USA) (Appendix A). The absorbance was measured at 450 nm using the Spectrostar Nanospectrophotometer (BMG Labtech, Ortenberg, Germany).

#### 4.6.2. Liver Biochemical Analysis

Liver tissue samples and the homogenate medium used for determining liver triglycerides levels, superoxide dismutase, and glutathione peroxidase activity were prepared according to the manufacturer’s instructions (Elabscience Biotechnology Co., Ltd., Houston, TX, USA). Briefly, 20 mg of liver tissue was harvested for each assay. The tissue was washed with ice-cold phosphate-buffered saline (PBS, 0.01 M, pH 7.4) and then homogenized on ice in 180 µL of PBS. The homogenate was centrifuged at 10,000× *g* for 10 min at 4 °C. The supernatant was collected and kept on ice for immediate analysis or stored at −80 °C for later use.

Liver triglyceride (TAG) levels were quantified using the triglyceride colorimetric assay kit (GPO-PAP method, catalog no.: E-BC-K238) according to the manufacturer’s instructions (Elabscience Biotechnology Co., Ltd., Houston, TX, USA) (Appendix A). The activity of superoxide dismutase (SOD) was determined using the total superoxide dismutase (T-SOD) activity assay kit (WST-1 method, catalog no.: E-BC-K020-M) according to the manufacturer’s instructions (Appendix A). The activity of glutathione peroxidase (Gpx) was measured using the glutathione peroxidase (GSH-Px) activity assay kit (catalog no.: E-BC-K096-S) (Appendix A). The Spectrostar Nanospectrophotometer (BMG Labtech, Ortenberg, Germany) was used to measure the absorbance for TAG at 510 nm, SOD at 450 nm, and GPx at 412 nm.

### 4.7. MDA Assay

The liver malondialdehyde (MDA) concentration was quantified following a previously established laboratory protocol [82]. Briefly, 50 mg of liver tissue was homogenized in 500 µL of 0.2% phosphoric acid. The homogenate was centrifuged at 400× *g* for 10 min. Thereafter, 400 µL of the homogenate was combined with 400 µL of 2% phosphoric acid and then divided into two glass tubes, each receiving equal volumes of the solution. Then, 200 µL of 7% phosphoric acid was added to both tubes, followed by the addition of 400 µL of thiobarbituric acid (TBA)/butylated hydroxytoluene (BHT) solution to the sample tube and 400 µL of 3 mM hydrochloric acid (HCl) to the blank tube. To adjust the pH to 1.5, 200 µL of 1 M HCl was added to both tubes. The tubes were heated at 100 °C for 15 min and allowed to cool to room temperature. After cooling, 1.5 mL of butanol was added, the solution was vortexed for 1 min for thorough mixing, and then allowed to settle until two phases could be distinguished. The upper butanol phase was transferred to Eppendorf tubes and centrifuged at 13,200× *g* for 6 min. The samples were aliquoted into a 96-well microtiter plate in triplicate, and the absorbance was measured at 532 nm (reference 600 nm) using a Spectrostar Nanospectrophotometer (BMG Labtech, Ortenberg, Germany). The absorbance from these wavelengths was used to calculate the concentration of MDA using Beer’s Law as follows: MDA concentration = Average Absorbance/Absorption coefficient (156 nm) [82].

### 4.8. Histological Analysis

A histological examination of the liver was performed on two randomly selected animals per group, following a previously documented protocol [83]. Following removal from the animals, liver tissues were immediately fixed in 10% formalin buffer. The samples were dehydrated with alcohol and embedded in paraffin wax. A microtome was used to slice 5 µm sections of liver tissue, which were then stained with hematoxylin and eosin (H&E). The processed tissue sections were visualized and captured with a Leica SCN400 scanner and analyzed using Aperio ImageScope software (version 12.4.6; Leica Biosystems, Buffalo Grove, IL, USA).

Lipid droplet scores were assigned using the Brunt method [84] as follows: Grade 0: none; Grade 1: <33% hepatocytes affected; Grade 2: 33–66% hepatocytes affected; Grade 3: >66% hepatocytes affected.

Nuclei displacement scores were assigned as follows: Grade 0: none; Grade 1: 1–24% displaced nuclei; Grade 2: 25–50% displaced nuclei; Grade 3: >50% displaced nuclei.

Sinusoid disruption scores were assigned as follows: Grade 0: normal well-defined sinusoids; Grade 1: mild changes with some sinusoidal dilation but no significant loss; Grade 2: moderate disruption with sinusoidal dilation and focal areas of loss; Grade 3: severe disruption with large loss of sinusoidal space.

Hepatocyte plate arrangement scores were assigned as follows: Grade 0: radial plate arrangement; Grade 1: mild disruption where some plates are irregular but general organization is maintained; Grade 2: moderate disruption where plates are disorganized with gaps; Grade 3: severe disruption with no clear plate formation.

Binuclear hepatocyte scores were assigned as follows: Grade 0: none; Grade 1: <5% binucleated hepatocytes; Grade 2: 5–20% binucleated hepatocytes; Grade 3: >20% binucleated hepatocytes.

### 4.9. Statistical Analysis

Data were presented as the mean ± standard error of the mean (SEM). The Shapiro–Wilk test was used to evaluate the normality of the data and Bartlett’s test was used to test for homogeneity of variance. A one-way analysis of variance (ANOVA) was used to compare differences among all groups for the dependent variable. An ordinary two-way ANOVA was conducted on the GTIN+ND, GTIN+HFHC, MET+ND, and MET+HFHC groups. This analysis assessed the effects of two factors, namely diet (ND vs. HFHC) and treatment (GTIN vs. MET) on the dependent variable. Post hoc comparisons were made using the Tukey test via GraphPad Prism 5 software. Statistical significance was determined at a value of *p* < 0.05.

## 5. Limitations and Future Recommendations

A limitation of this study was the restricted sample size for histological analysis, as only two animals per group were examined due to logistical constraints and shortages of laboratory supplies. Additionally, while lipid peroxidation and antioxidant activity were assessed, only selected markers were measured. Future studies should investigate a broader range of oxidative stress markers and antioxidant enzymes. Quantitative histological analysis should also be conducted on a larger sample size, and control animals without DMSO should be included in future studies.

## 6. Conclusions

Taken together, GTIN may serve as an effective candidate for managing liver complications in the pre-diabetic state in the presence and absence of dietary modification. This was evidenced by reduced liver TAGs, liver weights, and plasma SREBP-1c levels in the GTIN-treated animals. It also reduced liver MDA levels and injury while improving plasma bilirubin levels, SOD activity, Gpx activity, and liver histology. Similar results were observed for the liver parameters in the MET-treated groups. The findings of this study may suggest that GTIN and MET have beneficial effects on liver health in diet-induced pre-diabetes, with the best outcomes achieved when treatment and dietary intervention are used in conjugation. Notably, GTIN offers greater protective effects in comparison to MET, particularly in rats without dietary intervention.

## Figures and Tables

**Figure 1 molecules-30-01834-f001:**
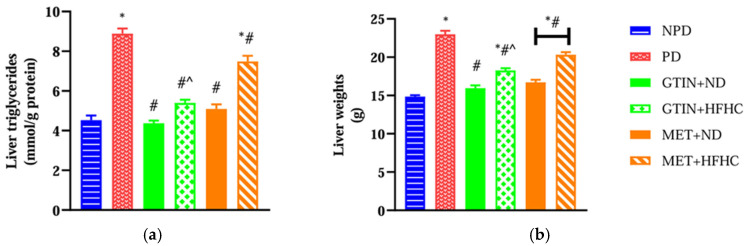
The effects of GTIN on (**a**) liver TAG levels and (**b**) liver weights in rats in both the presence and absence of dietary intervention at week 32 of the experimental period (*n* = 6, per group). Values are presented as mean ± SEM. * = *p* < 0.05 denotes comparison with the non-pre-diabetic control (NPD); # = *p* < 0.05 denotes comparison with the pre-diabetic control (PD); ^ = *p* < 0.05 denotes comparison with the metformin + high-fat, high-carbohydrate (MET+HFHC) group.

**Figure 2 molecules-30-01834-f002:**
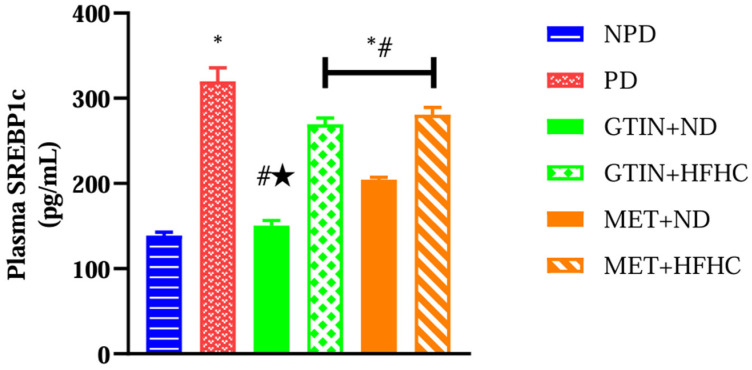
The effects of GTIN on plasma SREBP-1c levels in rats in both the presence and absence of dietary intervention at week 32 of the experimental period (*n* = 6, per group). Values are presented as mean ± SEM. * = *p* < 0.05 denotes comparison with the non-pre-diabetic control (NPD); # = *p* < 0.05 denotes comparison with the pre-diabetic control (PD); ★ = *p* < 0.05 denotes comparison with the metformin + normal diet (MET+ND) group.

**Figure 3 molecules-30-01834-f003:**
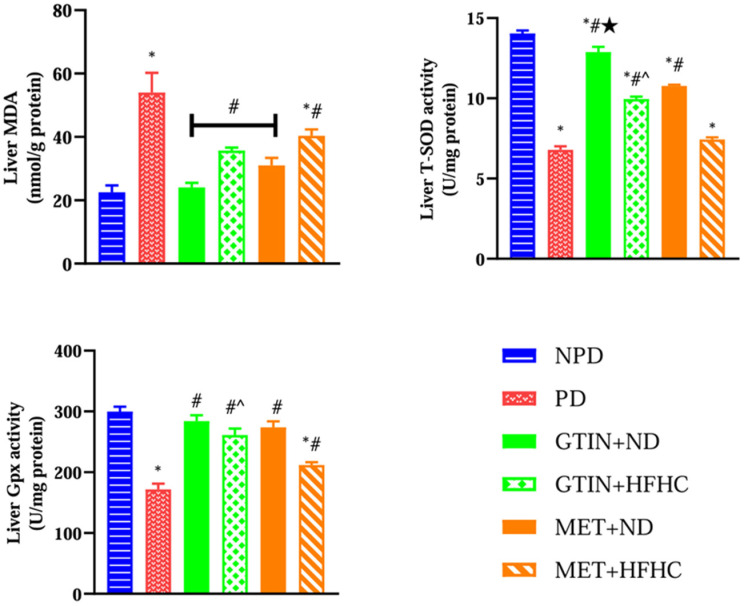
The effects of GTIN on liver oxidative status in rats in both the presence and absence of dietary intervention at week 32 of the experimental period (*n* = 6, per group). Values are presented as mean ± SEM. * = *p* < 0.05 denotes comparison with the non-pre-diabetic control (NPD); # = *p* < 0.05 denotes comparison with the pre-diabetic control (PD); ★ = *p* < 0.05 denotes comparison with the metformin + normal diet (MET+ND) group; ^ = *p* < 0.05 denotes comparison with the metformin+ high-fat, high-carbohydrate (MET+HFHC) group.

**Figure 4 molecules-30-01834-f004:**
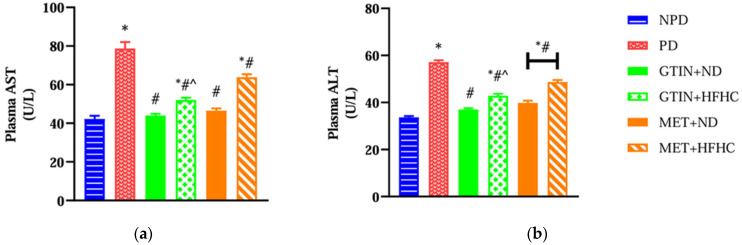
The effects of GTIN on plasma (**a**) AST and (**b**) ALT levels in rats in both the presence and absence of dietary intervention at week 32 of the experimental period (*n* = 6, per group). Values are presented as mean ± SEM. * = *p* < 0.05 denotes comparison with the non-pre-diabetic control (NPD); # = *p* < 0.05 denotes comparison with the pre-diabetic control (PD); ^ = *p* < 0.05 denotes comparison with the metformin + high-fat, high-carbohydrate (MET+HFHC) group.

**Figure 5 molecules-30-01834-f005:**
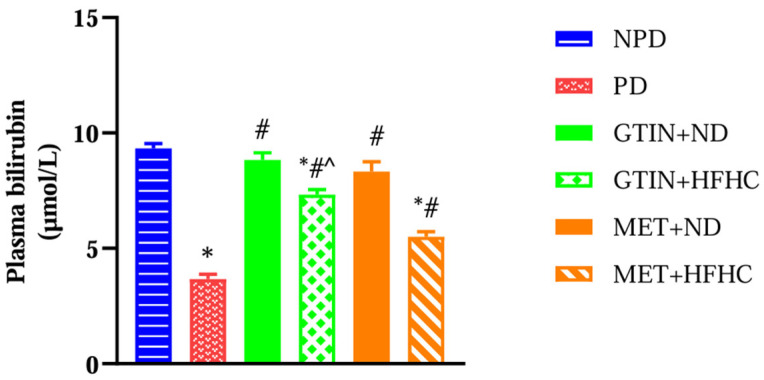
The effects of GTIN on plasma bilirubin levels in rats in both the presence and absence of dietary intervention at week 32 of the experimental period (*n* = 6, per group). Values are presented as mean ± SEM. * = *p* < 0.05 denotes comparison with the non-pre-diabetic control (NPD); # = *p* < 0.05 denotes comparison with the pre-diabetic control (PD); ^ = *p* < 0.05 denotes comparison with the metformin + high-fat, high-carbohydrate (MET+HFHC) group.

**Figure 6 molecules-30-01834-f006:**
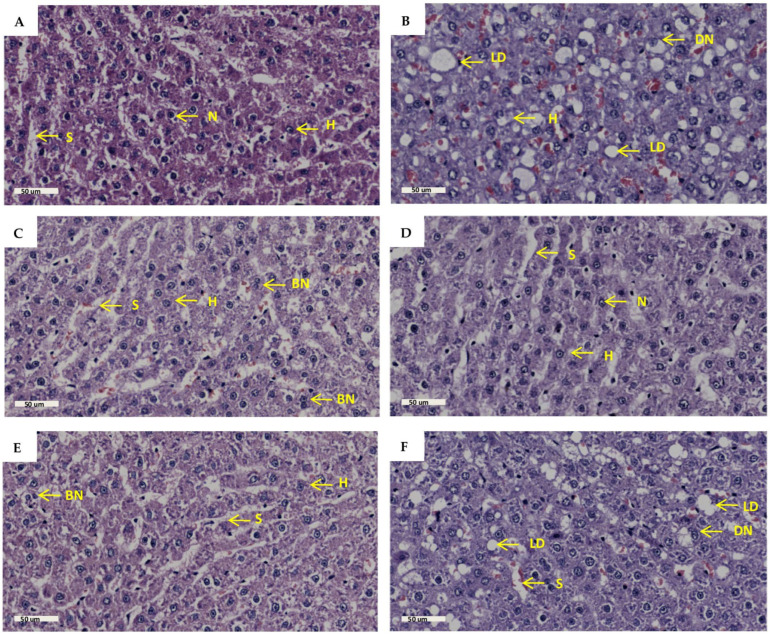
The effects of GTIN on liver histopathology in rats in both the presence and absence of dietary intervention at week 32 of the experimental period (*n* = 2, per group). Hematoxylin and eosin stain (H&E), magnification 40×, scale bar length = 50 µm. (**A**) NPD group; (**B**) PD group; (**C**) GTIN+ND group; (**D**) GTIN+HFHC group; (**E**) MET+ND group; (**F**) MET+HFHC group. BN = binuclear; DN= displaced nucleus; H = hepatocyte; LD = lipid droplet; N= nucleus; S = sinusoid.

**Figure 7 molecules-30-01834-f007:**
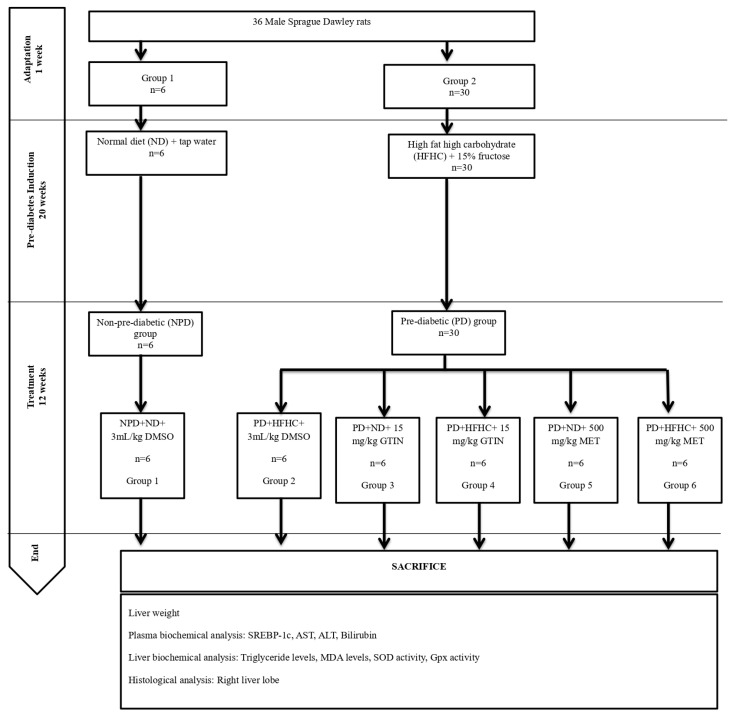
Layout of experimental design and grouping.

**Table 1 molecules-30-01834-t001:** Histopathological scoring of liver tissue at week 32 of the experimental period (*n* = 2, per group).

Groups	Lipid Droplets	Nuclei Displacement	Sinusoid Arrangement	Hepatocyte Plate Arrangement	Binuclear Hepatocytes
NPD	0	0	0	0	1
PD	3	2	3	3	1
GTIN+ND	1	1	1	1	1
GTIN+HFHC	1	1	1	1	1
MET+ND	1	1	1	1	1
MET+HFHC	2	1	2	2	1

## Data Availability

The data presented in this study are available upon request from the corresponding author.

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
