# Peer review of "Investigating the Effects of Gossypetin on Liver Health in Diet-Induced Pre-Diabetic Male Sprague Dawley Rats"

_molecules, 2025, doi:10.3390/molecules30081834_

Round 1
Reviewer 1 Report
Comments and Suggestions for Authors
Please see the attacment.

Reviewer 2 Report
Comments and Suggestions for Authors
This manuscript describes the effect of gossypetin on liver health in diet-induced pre-diabetic male Sprague Dawley rats. The Authors examine the unexplored effect of this quercetin analog on liver health in pre-diabetic state. In my opinion the topic is relevant to the field as long-term diet and excersise are burdensome for the patient. Since now, there are no report examining this problem. The experiment were well planned however the Authors have to unify the units. The conclusions drawn from the research are correctly edited. The references have to be refreshed because there are 39 references from the years 200-2015. The Authors should unified the units in the whole work. They used for example mg/Kg, kcal/g what is incorrect.
Reviewer 3 Report
Comments and Suggestions for Authors
Peer review report on ‘Investigating the effects of gossypetin on liver health in diet-induced pre-diabetic male Sprague Dawley rats’.
Manuscript ID: Molecules-3502624
This paper describes gossypetin, a flavonoid extracted from the calyx and flowers of the shrub Hibiscus sabdariffa, and its health benefits with regards to the pre-diabetic liver. Gossypetin has previously been found to demonstrate interesting antioxidant, anti-inflammatory and anti-cancer properties and to have a positive influence on glucose homeostasis and cardiovascular health in the pre-diabetic state. However, as the liver plays such a key role in diabetes mellitus this research focuses on this issue.
The manuscript is well researched with appropriate experimental methods, and the illustrations are clear and informative. The references are appropriate. The English is excellent, the work is well-described, and the paper has been meticulously prepared for submission which is very pleasant to encounter. The results are convincing.
My only comment is that in the results section, Section 2, it would be helpful to add to each experiment a brief explanation for the reason why you are investigating this factor. This leads into the discussion which then expands on this synopsis and interprets the results.
Round 2
Reviewer 1 Report
Comments and Suggestions for Authors
Please see the attachment.
